# Public knowledge, attitudes and practices towards COVID-19: A cross-sectional study in Malaysia

Arina Anis Azlan[1], Mohammad Rezal Hamzah[2], Tham Jen Sern[3], Suffian Hadi Ayub[4], Emma Mohamad[1]*

1 Centre for Research in Media and Communication, Faculty of Social Sciences and Humanities, Universiti Kebangsaan Malaysia, Bangi, Selangor, Malaysia, 2 School of Human Development and Technocommunication, Universiti Malaysia Perlis, Arau, Perlis, Malaysia, 3 Department of Communication, Faculty of Modern Languages and Communication, Universiti Putra Malaysia, Seri Kembangan, Selangor, Malaysia, 4 Department of Communication, School of Arts, Sunway University, Petaling Jaya, Selangor, Malaysia

* emmamohamad@ukm.edu.my

**Data Availability Statement:** All relevant data are within the manuscript and its Supporting Information files.

## Abstract

In an effort to mitigate the outbreak of COVID-19, many countries have imposed drastic lockdown, movement control or shelter in place orders on their residents. The effectiveness of these mitigation measures is highly dependent on cooperation and compliance of all members of society. The knowledge, attitudes and practices people hold toward the disease play an integral role in determining a society's readiness to accept behavioural change measures from health authorities. The aim of this study was to determine the knowledge levels, attitudes and practices toward COVID-19 among the Malaysian public. A cross-sectional online survey of 4,850 Malaysian residents was conducted between 27th March and 3rd April 2020. The survey instrument consisted of demographic characteristics, 13 items on knowledge, 3 items on attitudes and 3 items on practices, modified from a previously published questionnaire on COVID-19. Descriptive statistics, chi-square tests, t-tests and one-way analysis of variance (ANOVA) were conducted. The overall correct rate of the knowledge questionnaire was 80.5%. Most participants held positive attitudes toward the successful control of COVID-19 (83.1%), the ability of Malaysia to conquer the disease (95.9%) and the way the Malaysian government was handling the crisis (89.9%). Most participants were also taking precautions such as avoiding crowds (83.4%) and practising proper hand hygiene (87.8%) in the week before the movement control order started. However, the wearing of face masks was less common (51.2%). This survey is among the first to assess knowledge, attitudes and practice in response to the COVID-19 pandemic in Malaysia. The results highlight the importance of consistent messaging from health authorities and the government as well as the need for tailored health education programs to improve levels of knowledge, attitudes and practices.

**Funding:** This research was supported by a grant from Universiti Kebangsaan Malaysia (Grant code: SK-2020-007) to EM. The funder had no role in the study design, data collection and analysis, decision to publish, or preparation of the manuscript.

**Competing interests:** The authors have declared that no competing interests exist.

# 1. Introduction

The coronavirus disease 2019 (COVID-19) emerged in Wuhan, China at the end of 2019. Since then, it has spread to 200 countries and has been declared a global pandemic by the World Health Organisation (WHO). To date, there are more than 2.3 million positive COVID-19 cases recorded with at least 150,000 deaths globally [1].

The first case of COVID-19 in Malaysia was detected on 25th January 2020 involving three tourists from China [2]. The number of cases steadily increased before the nation's first two deaths were recorded on 17th March [3]. As of 20th April 2020, Malaysia has recorded more than 5300 positive cases involving 89 deaths. A majority of these cases were traced back to a religious gathering which has now reached its fifth-generation infections [4]. The Malaysian Prime Minister enforced a movement control order (MCO) on 18th March 2020 as a mitigation effort to reduce community spread and the overburdening of the country's health system. Similar to lockdowns in China and Italy, the MCO restricted most non-essential activity outside the home. Malaysians were only permitted to leave the house for basic activities such as buying groceries and seeking medical treatment. The MCO also restricted Malaysians from leaving the country and all foreigners from entry. Non-essential sectors were ordered to close operations or allow employees to work from home.

Lockdown measures were perceived as necessary to curb the spread of the virus as rapid human-to-human transmission occurred and much about the virus remained unknown [5]. Due to the obscurity of this novel virus, there has been a lot of confusion and misunderstanding about the virus itself, how it can spread and the necessary precautions that should be taken to prevent infection. This becomes increasingly challenging with the vast amount of misinformation and disinformation shared on social media that is clouding people's understanding of COVID-19 [6]. When the initial MCO announcement was made, Malaysians reacted in panic and confusion. Aside from panic buying, people crowded public transportation hubs to travel back to their hometowns, potentially increasing the risk of infection to other parts of the country. While this reaction to the MCO was not unexpected, it raises questions regarding the level of understanding and attitudes toward COVID-19 among Malaysians.

The knowledge, attitudes and practices (KAP) toward COVID-19 play an integral role in determining a society's readiness to accept behavioural change measures from health authorities. KAP studies provide baseline information to determine the type of intervention that may be required to change misconceptions about the virus. Assessing the KAP related to COVID-19 among the general public would be helpful to provide better insight to address poor knowledge about the disease and the development of preventive strategies and health promotion programs. Among the lessons learned from the SARS outbreak is that knowledge and attitudes are associated with levels of panic and emotion which could further complicate measures to contain the spread of the disease [7,8]. The survey also gives a general picture of Malaysians COVID-19 prevention practices before the MCO and this can better prepare the government to address future health crises involving infectious diseases. The results of this study are important to inform future efforts focusing on societal readiness to comply with pandemic control measures.

# 2. Methods

## Study design

A quantitative approach was utilised to achieve the objectives of this study. A survey is most appropriate as it allows large populations to be assessed with relative ease [9]. In this study, a cross-sectional survey was deemed most appropriate to gather information on COVID-19 for

the Malaysian context. Data collection was performed online using the Survey Monkey platform. The call for participation was made on social media.

## Ethical approval

The Ethics Committee of Universiti Kebangsaan Malaysia approved our study protocol, procedures, information sheet and consent statement (JEP-2020-276). Participants who gave consent to willingly participate in the survey would click the 'Continue' button and would then be directed to complete the self-administered questionnaire.

## Recruitment procedure

This cross-sectional survey was conducted in the second week of the MCO, between 27[th] March 2020 to 3[rd] April 2020. The target sample size was 3,640, determined by identifying the smallest acceptable size of a demographic subgroup with a ±5% margin of error and a confidence level of 95% [10,11]. As it was not feasible to conduct a systematic nationwide sampling procedure during this period, the researchers opted to use an online survey using Survey Monkey Advantage Annual. Members of the Malaysian public above the age of 18 and currently residing in the country were eligible to participate in the survey. We utilised several strategies to reach as many respondents as possible all over the country within the one-week data collection period. This includes relying on professional and personal networks of the researchers, reaching out to community leaders and social media influencers to broadcast and share the survey. Two main platforms used in disseminating this survey were social media (Facebook, Twitter and Instagram) and WhatsApp. Facebook and Whatsapp were selected as two of the most popular communication and social platforms in Malaysia [12]. While Facebook is generally preferred by older Malaysians, Twitter and Instagram are more popular among the younger generation. A standardised general description about the survey was given in the WhatsApp message/social media postings before the link was provided to both English and Malay language versions of the questionnaire. A total of 4,850 participants took part in the survey.

## Study instrument

The survey instrument is an adaptation of the measures developed in a study on Chinese residents' knowledge, attitudes and practices (KAP) towards COVID-19 in China [13]. The questionnaire consisted of four main themes: 1) demographics, which surveyed participants' sociodemographic information, including gender, age, state of residence, occupation, and household income; 2) knowledge about COVID-19; 3) attitudes toward COVID-19; and 4) practices relevant to COVID-19. The survey was offered in the English and Malay languages. A backward-translation approach was used in translating the items between English and Bahasa Malaysia, so as to ensure linguistic and conceptual equivalence [14]. Discrepancies between the two versions were rectified, and equivalence of measuring on all items was ensured through consultation with bilingual researchers.

To measure knowledge about COVID-19, 13 items were adapted from previous research [13]. These items include the participant knowledge about clinical presentations (items 1–4), transmission routes (items 5–8) and prevention and control (items 9–13) of COVID-19. Participants were given "true," "false," or "not sure" response options to these items. A correct response to an item was assigned 1 point, while an incorrect/not sure response was assigned 0 points. The maximum total score ranged from 0–13, with a higher score indicating better knowledge about COVID-19.

To measure attitudes towards COVID-19, surveyed participants were asked whether they agreed, disagreed or were not sure that the pandemic would be successfully controlled. They were also asked about their confidence towards the government in winning the battle against COVID-19 (yes or no) and about the ability of the government in handling the COVID-19 crisis (agree, disagree, or not sure). To measure practices, participants were asked yes/no questions on whether they had avoided going to crowded places such as weddings; wore a face mask when leaving home; and whether they practiced proper hand hygiene in the week before the movement control order (MCO).

## Statistical analysis

For this study, the collected data were analysed using the Statistical Package for the Social Sciences (SPSS), version 26. Descriptive analysis focused on frequencies, and percentages while chi-square tests, independent samples t-tests and one-way analysis of variance (ANOVA) were utilised to determine the differences between groups for selected demographic variables. The statistical significance level was set at $p < 0.05$. Internal consistency of the knowledge measures was tested using a reliability test where the Cronbach alpha coefficient aided in determining the reliability of the variables. The results showed that the Cronbach alpha for knowledge (13 items) was 0.655. The result added credence where according to Griethuijsen, the range of Cronbach alpha within 0.6 to 07 is considered adequate and reliable [15]. It is attested that the items used to measure knowledge on COVID-19 are therefore acceptable.

## 3. Results

### Demographic characteristics

A total of 4850 participants participated in the study. Out of the total, the average age was 34 years (SD = 11.2, range = 18–73), 2808 (57.9%) were women, 1993 (41.1%) resided in Central Malaysia and 2173 (44.8%) were employed in the public sector. Other demographic characteristics are detailed in Table 1.

### Assessment of knowledge

A total of thirteen questions were used to measure knowledge on the COVID-19 virus. The average knowledge score for participants was 10.5 (SD = 1.4, range 0–13). The overall correct answer rate of the knowledge questionnaire was 80.5% (10.5/13*100) while the range of correct answer rates for all participants were between 46.2 to 100%. About 77.2% of participants were able to obtain scores above 10, representing an acceptable level of knowledge on COVID-19.

Most participants knew that people who had contact with an infected person should be immediately isolated for a period of 14 days (99.1%) and that this is an effective way to reduce the spread of the virus (98.9%). Even so, there was noticeable confusion among participants regarding transmission of the virus. Only 43.3% of participants answered correctly when asked if the virus was airborne and just 35.7% answered correctly when asked if eating and touching wild animals could result in infection [Table 2].

Differences in knowledge scores among different demographic characteristics were assessed using t-tests and ANOVA. The results show that knowledge scores were significantly different across genders, age groups, regions, occupation groups and income categories. Higher knowledge scores were obtained among female participants, those above the age of 50, people residing in Central Malaysia and in the higher income category.

The results of the ANOVA analyses show that the knowledge scores of people living in the Central region were significantly higher than other regions. Additionally, the average

**Table 1. Demographic characteristics of participants (N = 4850).**

| Characteristic | | Number | Percentage (%) |
|---|---|---|---|
| Gender | Male | 2042 | 42.1 |
| | Female | 2808 | 57.9 |
| Age | 18–29 | 2065 | 42.6 |
| | 30–49 | 2233 | 46.1 |
| | Above 50 | 544 | 11.2 |
| Region | Central | 1993 | 41.1 |
| | Northern | 1178 | 24.3 |
| | Southern | 638 | 13.2 |
| | Eastern | 662 | 13.6 |
| | Sabah/Sarawak | 379 | 7.8 |
| Occupation | Public sector | 2173 | 44.8 |
| | Student | 1125 | 23.2 |
| | Private sector | 955 | 19.7 |
| | Self-employed | 267 | 5.5 |
| | Not employed | 195 | 4.0 |
| | Retiree | 96 | 2.0 |
| | Other* | 32 | 0.7 |
| Income category | Under RM 3,000 per month | 1540 | 31.8 |
| | RM3,001 –RM6,000 per month | 1289 | 26.6 |
| | RM6,001 –RM9,000 per month | 832 | 17.2 |
| | RM9,001 –RM12,000 per month | 575 | 11.9 |
| | RM12,001 and above per month | 614 | 12.7 |

* "Other" includes occupations such as manual labour and contract/ part-time work

knowledge score of students were significantly lower than those of other occupation categories and those earning below RM3,000 per month showed significantly lower knowledge scores [Table 3].

## Assessment of attitudes

Participants were asked three questions in assessment of attitudes. The first question asked whether or not they agreed that the COVID-19 situation would be successfully controlled; second, whether they thought Malaysia would be able to win its battle against the virus; and third, whether they thought the Malaysian government was handling the health crisis well [Fig 1].

For the first question, a majority of participants agreed that COVID-19 would successfully be controlled (83.1%). Even so, 14% of participants were unsure whether the virus would be controlled and a smaller number of participants disagreed that it would be successfully controlled (2.1%). The attitude of successfully controlling COVID-19 was significantly associated with age group, region and occupation. Knowledge scores of those who were unsure were also significantly lower than those who agreed that the virus would be successfully controlled [Table 4].

For the second attitude question, the majority of participants had confidence that Malaysia would be able to win the battle against COVID-19 (95.9%), while a small percentage did not have that confidence (3.3%). The confidence that Malaysia would be able to win the battle against COVID-19 was associated with age group and occupation. No significant difference was found between the two confidence groups in terms of knowledge score.

**Table 2. Participant knowledge of COVID-19 (N = 4850).**

| Question | True | False | I'm not sure |
|---|---|---|---|
| 1. The main clinical symptoms of COVID-19 are fever, fatigue, dry cough, and body aches. | **4207 (86.7%)** | 490 (10.1%) | 153 (3.2%) |
| 2. Unlike the common cold, stuffy nose, runny nose, and sneezing are less common in persons infected with the COVID-19 virus. | **3152 (65.0%)** | 862 (17.8%) | 836 (17.2%) |
| 3. There currently is no effective cure for COVID-19, but early symptomatic and supportive treatment can help most patients recover from the infection. | **4562 (94.1%)** | 66 (1.4%) | 222 (4.6%) |
| 4. Not all persons with COVID-2019 will develop to severe cases. Only those who are elderly and have chronic illnesses are more likely to be severe cases. | **4347 (89.6%)** | 277 (5.7%) | 226 (4.7%) |
| 5. Eating or touching wild animals would result in the infection by the COVID-19 virus. | 1705 (35.2%) | **1731 (35.7%)** | 1414 (29.2%) |
| 6. Persons with COVID-19 cannot infect the virus to others if they do not have a fever. | 281 (5.8%) | **4038 (83.2%)** | 535 (11.0%) |
| 7. The COVID-19 virus spreads via respiratory droplets of infected individuals. | **3971 (81.9%)** | 359 (7.4%) | 520 (10.7%) |
| 8. The COVID-19 virus is airborne. | 2042 (42.1%) | **2099 (43.3%)** | 709 (14.6%) |
| 9. Ordinary residents can wear face masks to prevent the infection by the COVID-19 virus. | **3719 (76.7%)** | 813 (16.8%) | 318 (6.6%) |
| 10. It is not necessary for children and young adults to take measures to prevent the infection by the COVID-19 virus. | 179 (3.7%) | **4630 (95.5%)** | 41 (0.8%) |
| 11. To prevent the infection by COVID-19, individuals should avoid going to crowded places and avoid taking public transportations. | **4689 (96.7%)** | 112 (2.3%) | 49 (1.0%) |
| 12. Isolation and treatment of people who are infected with the COVID-19 virus are effective ways to reduce the spread of the virus. | **4797 (98.9%)** | 17 (0.4%) | 36 (0.7%) |
| 13. People who have contact with someone infected with the COVID-19 virus should be immediately isolated in a proper place. In general, the isolation period is 14 days. | **4807 (99.1%)** | 18 (0.4%) | 25 (0.5%) |

Correct answers are indicated in bold.

The third attitude question asked whether the participant agreed that the Malaysian government was handling the COVID-19 health crisis well. A large percentage of participants agreed with this statement (89.9%). Rates of disagreement and uncertainty were at 3.8% and 5.4% respectively. Agreement that the Malaysian government was performing well in handling the COVID-19 crisis was significantly associated with gender, age group, region and occupation. Knowledge scores were also significantly different between those who agreed that the government was doing a good job at handling the crisis and those who were unsure.

## Assessment of practices

Practices toward COVID-19 were measured using three questions enquiring on: 1) avoidance of crowded places, 2) wearing of face masks; and 3) practising proper hand hygiene in the week before the Movement Control Order (MCO) was implemented in Malaysia [Fig 2].

For the first question, 83.4% of participants reported that they were avoiding crowded places in the week before the MCO was implemented. The other 16.6% did not avoid crowded places.

In examining the differences between demographic groups, it was found that there were significant associations between age group, income category and avoidance of crowded places. Younger people and those earning below RM3,000 monthly were more avoidant of crowded places in the week before the MCO. There were also significant differences in knowledge

**Table 3. Demographic characteristics of participants and knowledge score (N = 4850).**

| Characteristics | | No. of participants | Knowledge score (SD) | t / F | P |
|---|---|---|---|---|---|
| Gender | Male | 2042 (42.1%) | 10.3 (1.5) | -6.878 | <0.001 |
| | Female | 2808 (57.9%) | 10.6 (1.3) | | |
| Age group | 18–29 | 2065 (42.6%) | 10.0 (1.4) | 203.717 | <0.001 |
| | 30–49 | 2233 (46.1%) | 10.7 (1.3) | | |
| | Above 50 | 544 (11.2%) | 11.0 (1.1) | | |
| Region | Central | 1993 (41.1%) | 10.7 (1.3) | 31.548 | <0.001 |
| | Northern | 1178 (24.3%) | 10.1 (1.4) | | |
| | Southern | 638 (13.2%) | 10.4 (1.5) | | |
| | Eastern | 662 (13.6%) | 10.4 (1.4) | | |
| | Sabah/Sarawak | 379 (7.8%) | 10.4 (1.4) | | |
| Occupation group | Other* | 33 (0.7%) | 10.5 (1.2) | 21.349 | <0.001 |
| | Student | 1125 (23.2%) | 10.1 (1.4) | | |
| | Unemployed | 195 (4.0%) | 10.7 (1.2) | | |
| | Retiree | 96 (2.0%) | 10.9 (1.1) | | |
| | Private sector | 955 (19.7%) | 10.7 (1.3) | | |
| | Public sector | 2173 (44.8%) | 10.5 (1.4) | | |
| | Self-employed | 267 (5.5%) | 10.7 (1.4) | | |
| Income category | Below RM3k | 1540 (31.8%) | 10 (1.4) | 96.113 | <0.001 |
| | RM3k – 6k | 1289 (26.6%) | 10.5 (1.4) | | |
| | RM6k – 9k | 832 (17.2%) | 10.7 (1.3) | | |
| | RM9k – 12k | 575 (11.9%) | 10.8 (1.3) | | |
| | RM12k and above | 614 (12.7%) | 11.0 (1.2) | | |

*"Other" includes occupations such as manual labour and contract/ part-time work

scores between those who did and did not avoid crowded places. Those with higher knowledge scores did not avoid crowded places in the week before the MCO [Table 5].

The second question asked participants if they were wearing face masks when outside the home during the week before the MCO began. More than half of the participants reported wearing a face mask when going out in public (51.2%). The remaining participants did not wear a mask (48.8%).

The wearing of face masks was found to be significantly associated with gender, age group, region, occupation and income group. Males, people between the ages of 18 and 49, students and those earning less than RM3,000 per month showed higher percentages in wearing face masks when leaving the house. People living in the Central region, those above the age of 50 and people with an income over RM12,000 per month were less likely to wear a face mask. The results also show that there was a significant difference between knowledge scores in terms of mask-wearing. Those with higher knowledge scores did not wear masks when leaving the house in the week before the MCO was enforced.

Lastly, when enquired about hand hygiene, a majority of participants reported that they practised proper hand hygiene by frequently washing their hands and using hand sanitiser (87.8%). Even so, there was still a percentage of participants who were not practising proper hand hygiene in the week before the implementation of the MCO (12.2%).

There were significant associations found between proper hand hygiene and gender, age group, region and occupation. Females, those living in the Central region, people between the ages 18 to 29 and students were more likely to practise good hand hygiene. Those above 50,

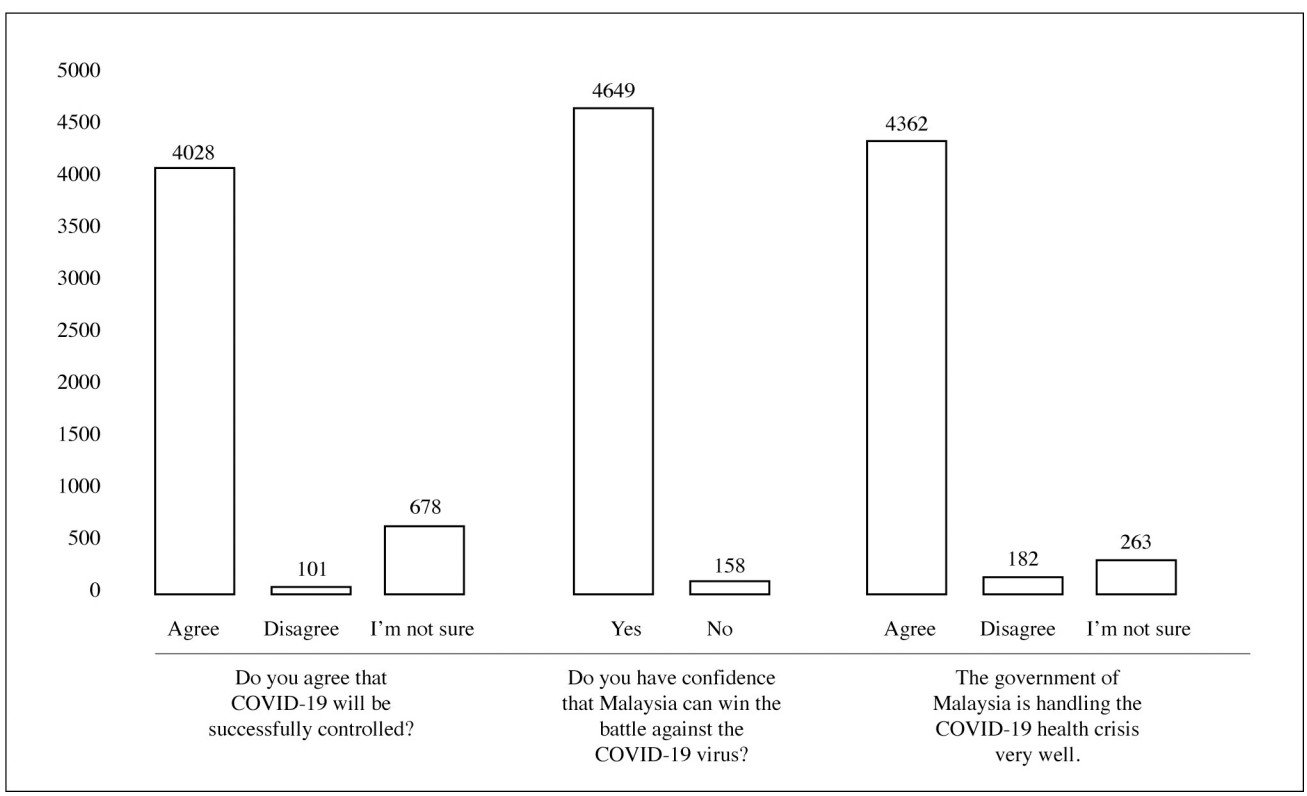

**Fig 1. Attitudes of participants on COVID-19 (N = 4850).**

residents in the Eastern region and retirees were among the highest percentage of participants who had not practised good hand hygiene in the week before the MCO.

## 4. Discussion

COVID-19 is a relatively new virus that has had devastating effects within the short time since it was first detected in December 2019. To date, there has been limited published data on population knowledge, attitudes and practices toward COVID-19, specifically in Malaysia. The novelty of this disease, along with its uncertainties, make it critical for health authorities to plan appropriate strategies to prepare and manage the public. It is therefore of utmost importance that the knowledge, attitudes and practices of the population be studied to guide these efforts.

The average knowledge score of Malaysians in regards to COVID-19 was moderate at 10.5 ±1.4 with an overall correct rate of 80.5%. Even so, correct rates of COVID-19 knowledge ranged widely indicating that while some participants had high levels of knowledge on the disease, others did not. Malaysians above the age of 50 held higher knowledge scores, possibly due to a higher risk perception of contraction and complications from the disease [16]. On the other hand, those with low monthly income scored among the lowest knowledge scores. This may indicate limited access to credible and timely information about the virus. This variation in levels of knowledge may be reflective of the current COVID-19 information landscape in the country. Although health authorities have been consistently disseminating COVID-19 information since the disease was first detected in Malaysia, there has also been a surge in false and inaccurate information [17,18]. The overload of information may have caused confusion and difficulty ascertaining correct information.

**Table 4. Demographic characteristics of participants and attitudes toward COVID-19 (N = 4850).**

| Characteristics | | Do you agree that COVID-19 will be successfully controlled? | | | Do you have confidence that Malaysia can win the battle against the COVID-19 virus? | | The government of Malaysia is handling the COVID-19 health crisis very well. | | |
|---|---|---|---|---|---|---|---|---|---|
| | | Agree | Disagree | I'm not sure | Yes | No | Agree | Disagree | I'm not sure |
| Gender | Male | 1724 (85.0%) | 44 (2.2%) | 259 (12.8%) | 1954 (96.4%) | 74 (3.6%) | 1836 (90.5%) | 94 (4.6%) | 98 (4.8%) |
| | Female | 2304 (82.9%) | 56 (2.0%) | 419 (15.1%) | 2695 (97.0%) | 84 (3.0%) | 2526 (90.9%) | 88 (3.2%) | 165 (5.9%)* |
| Age group | 18–29 | 1674 (81.6%) | 51 (2.5%) | 326 (15.9%) | 1967 (95.9%) | 84 (4.1%) | 1839 (89.7%) | 82 (4.0%) | 130 (6.3%) |
| | 30–49 | 1889 (85.3%) | 41 (1.9%) | 284 (12.8%) | 2160 (97.6%) | 54 (2.4%) | 2053 (92.7%) | 66 (3.0%) | 95 (4.3%) |
| | Above 50 | 459 (86%) | 8 (1.5%) | 67 (12.5%)** | 516 (96.6%) | 18 (3.4%)** | 464 (86.9%) | 34 (6.4%) | 36 (6.7%)*** |
| Region | Central | 1594 (80.8%) | 50 (2.5%) | 329 (16.7%) | 1899 (96.2%) | 74 (3.8%) | 1740 (88.2%) | 105 (5.3%) | 128 (6.5%) |
| | Northern | 975 (83.8%) | 21 (1.8%) | 167 (14.4%) | 1124 (96.6%) | 39 (3.4%) | 1066 (91.7%) | 33 (2.8%) | 64 (5.5%) |
| | Southern | 549 (86.5%) | 13 (2.0%) | 73 (11.5%) | 616 (97.0%) | 19 (3.0%) | 586 (92.3%) | 21 (3.3%) | 28 (4.4%) |
| | Eastern | 586 (88.9%) | 7 (1.1%) | 66 (10.0%) | 647 (98.2%) | 12 (1.8%) | 630 (95.6%) | 11 (1.7%) | 18 (2.7%) |
| | Sabah/Sarawak | 324 (85.9%) | 10 (2.7%) | 43 (11.4%)*** | 363 (96.3%) | 14 (3.7%) | 340 (90.2%) | 12 (3.2%) | 25 (6.6%)*** |
| Occupation group | Other# | 25 (78.1%) | 0 (0%) | 7 (21.9%) | 30 (93.8%) | 2 (6.3%) | 26 (81.3%) | 2 (6.3%) | 4 (12.5%) |
| | Student | 879 (78.6%) | 37 (3.3%) | 202 (18.1%) | 1061 (94.9%) | 57 (5.1%) | 974 (87.1%) | 59 (5.3%) | 85 (7.6%) |
| | Unemployed | 164 (85%) | 2 (1.0%) | 27 (14.0%) | 188 (97.4%) | 5 (2.6%) | 180 (93.3%) | 8 (4.1%) | 5 (2.6%) |
| | Retiree | 77 (81.9%) | 1 (1.1%) | 16 (17.0%) | 91 (96.8%) | 3 (3.2%) | 75 (79.8%) | 10 (10.6%) | 9 (9.6%) |
| | Private sector | 761 (80.3%) | 32 (3.4%) | 155 (16.4%) | 905 (95.5%) | 43 (4.5%) | 833 (87.9%) | 45 (4.7%) | 70 (7.4%) |
| | Public sector | 1889 (87.8%) | 28 (1.3%) | 234 (10.9%) | 2112 (98.2%) | 39 (1.8%) | 2042 (94.9%) | 44 (2.0%) | 65 (3.0%) |
| | Self-employed | 226 (85.6%) | 1 (0.4%) | 37 (14%)*** | 255 (96.6%) | 9 (3.4%)*** | 226 (85.6%) | 14 (5.3%) | 24 (9.1%)*** |
| Income category | Below RM3k | 1290 (84.3%) | 29 (1.9%) | 212 (13.8%) | 1486 (97.1%) | 45 (2.9%) | 1399 (91.4%) | 56 (3.7%) | 76 (5.0%) |
| | RM3k – 6k | 1093 (85.4%) | 26 (2.0%) | 161 (12.6%) | 1236 (96.6%) | 4 (3.4%) | 1154 (90.2%) | 41 (3.2%) | 85 (6.6%) |
| | RM6k – 9k | 698 (84.6%) | 14 (1.7%) | 113 (13.7%) | 795 (96.4%) | 30 (3.6%) | 749 (90.8%) | 37 (4.5%) | 39 (4.7%) |
| | RM9k – 12k | 456 (81.0%) | 12 (2.1%) | 95 (16.9%) | 545 (96.8%) | 18 (3.2%) | 519 (92.2%) | 19 (3.4%) | 25 (4.4%) |
| | RM12k and above | 491 (80.8%) | 20 (3.3%) | 97 (16.0%) | 587 (96.5%) | 21 (3.5%) | 541 (89.0%) | 29 (4.8%) | 38 (6.3%) |
| Knowledge score | | 10.5 (1.4) | 10.4 (1.4) | 10.2 (1.5)*** | 10.5 (1.4) | 10.3 (1.4) | 10.5 (1.4) | 10.3 (1.4) | 10.2 (1.5)*** |

#"Other" includes occupations such as manual labour and contract/ part-time work

*P<0.05

**P<0.01

***P<0.001

Several studies conducted in other Asian countries have indicated high levels of COVID-19 knowledge among the general population [13] and healthcare workers [19]. Differences in measurement and scoring systems do not make it possible for accurate comparisons of knowledge levels across these studies.

The present study found that a large majority of participants held positive attitudes toward overcoming COVID-19. Roughly eight out of ten participants agreed that COVID-19 would be successfully controlled. Similarly, approximately nine out of ten participants were confident that Malaysia would be able to win its battle against the virus and that the Malaysian government was handling the health crisis very well. High levels of positive attitudes were also detected in the KAP study conducted in China [13]. The authors attributed the positive attitudes to the drastic measures taken by the Chinese government in mitigating the spread of the virus. In Malaysia, the swift action taken by the Malaysian government in enforcing the MCO may have also contributed to these positive attitudes.

Although the percentage of participants reporting uncertainty toward success in fighting against COVID-19 was low (14%), this was significantly associated with lower knowledge

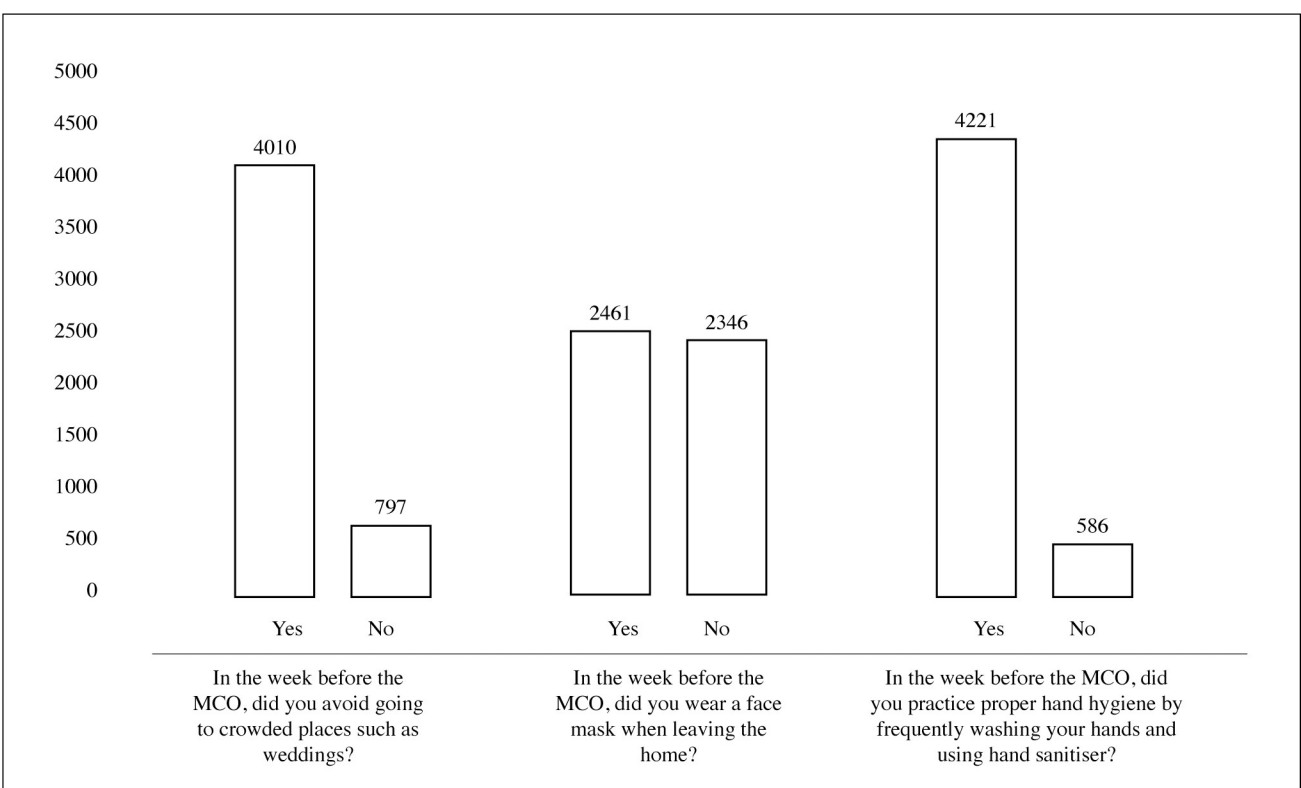

**Fig 2. Practices of participants on COVID-19 (N = 4850).**

scores. These results reinforce conclusions from previous studies associating higher levels of knowledge with higher confidence and positive attitudes in health crises [20].

Positive attitudes were higher among those working in the public sector. This group showed the highest confidence that COVID-19 would be successfully controlled, that Malaysia would win the battle against the disease and that the Malaysian government were handling the health crisis well. This may be due to work duties or affiliations directly related to government efforts toward the containment of the virus.

In the current study, most participants reported taking precautions such as avoiding crowded places and practising proper hand hygiene in the week before the MCO was implemented. This indicates a general willingness for participants to make behavioural changes in the face of the COVID-19 pandemic. Even so, people above the age of 50 and people who earned more than RM12,000 per month were among those who did not avoid crowded places in the week before the MCO. The week before the implementation of the MCO coincided with the school holidays in Malaysia, a common season for family holidays and gatherings such as weddings. Those above the age of 50 were also more likely to attend daily religious congregations like praying at the mosque. Cultural norms may have been influential in the decision to attend these gatherings despite the health risks, especially among the older generation. Previous research has also shown that those with higher income were less willing to comply with health recommendations [21] perceived less fear and more control in pandemic situations [22].

Interestingly, enquiry into the wearing of face masks garnered a mixed response. Almost half of the participants indicated that they did not wear a face mask when leaving the home in the week before the MCO. There are two possible explanations for this behaviour in the

**Table 5. Demographic characteristics of participants and practices toward COVID-19 (N = 4850).**

| Characteristics | | In the week before the MCO, did you avoid going to crowded places such as weddings? | | In the week before the MCO, did you wear a face mask when leaving the home? | | In the week before the MCO, did you practice proper hand hygiene by frequently washing your hands and using hand sanitizer? | |
|---|---|---|---|---|---|---|---|
| | | Yes | No | Yes | No | Yes | No |
| Gender | Male | 1701 (83.9%) | 327 (16.1%) | 1092 (53.8%) | 936 (46.2%) | 1701 (83.9%) | 327 (16.1%) |
| | Female | 2309 (83.1%) | 470 (16.9%) | 1369 (49.3%) | 1410 (50.7%)** | 2520 (90.7%) | 259 (9.3%)*** |
| Age group | 18–29 | 1742 (84.9%) | 309 (15.5%) | 1271 (62.0%) | 780 (38.0%) | 1849 (90.2%) | 202 (9.8%) |
| | 30–49 | 1853 (83.7%) | 361 (16.3%) | 972 (43.9%) | 1242 (56.1%) | 1932 (87.3%) | 282 (12.7%) |
| | Above 50 | 407 (76.2%) | 127 (23.8%)*** | 212 (39.7%) | 322 (60.3%)*** | 433 (81.1%) | 101(18.9%)*** |
| Region | Central | 1634 (82.8%) | 339 (17.2%) | 919 (46.6%) | 1054 (53.4%) | 1775 (90.0%) | 198 (10.0%) |
| | Northern | 984 (84.6%) | 179 (15.4%) | 695 (59.8%) | 468 (40.2%) | 1027 (88.3%) | 136 (11.7%) |
| | Southern | 533 (83.9%) | 102 (16.6%) | 318 (50.1%) | 317 (49.9%) | 546 (86.0%) | 89 (14.0%) |
| | Eastern | 535 (81.2%) | 124 (18.8%) | 323 (49.0%) | 336 (51.0%) | 553 (83.9%) | 106 (16.1%) |
| | Sabah/Sarawak | 324 (85.9%) | 53 (14.1%) | 206 (54.6%) | 171 (45.4%)*** | 320 (84.9%) | 57 (15.1%)*** |
| Occupation group | Other# | 26 (81.3%) | 6 (18.8%) | 12 (37.5%) | 20 (62.5%) | 24 (75.0%) | 8 (25.0%) |
| | Student | 927 (82.9%) | 191 (17.1%) | 697 (62.6%) | 421 (37.7%) | 1015 (90.8%) | 103 (9.2%) |
| | Unemployed | 161 (83.4%) | 32 (16.6%) | 81 (42.0%) | 112 (58.0%) | 168 (87.0%) | 25 (13.0%) |
| | Retiree | 73 (77.7%) | 21 (22.3%) | 34 (36.2%) | 60 (63.8%) | 71 (75.5%) | 23 (24.5%) |
| | Private sector | 780 (82.3%) | 168 (17.7%) | 429 (45.3%) | 519 (54.7%) | 851 (89.8%) | 97 (10.2%) |
| | Public sector | 1827 (84.9%) | 324 (15.1%) | 1086 (50.5%) | 1065 (49.5%) | 1858 (86.4%) | 293 (13.6%) |
| | Self-employed | 210 (79.5%) | 54 (20.5%) | 119 (45.1%) | 145 (54.9%)*** | 227 (86.0%) | 37 (14.0%)*** |
| Income category | Below RM3k | 1321 (86.3%) | 210 (13.7%) | 941 (61.5%) | 590 (38.5%) | 1368 (89.4%) | 163 (10.6%) |
| | RM3k – 6k | 1075 (84.0%) | 205 (16.0%) | 676 (52.8%) | 604 (47.2%) | 1115 (87.1%) | 165 (12.9%) |
| | RM6k – 9k | 682 (82.7%) | 143 (17.3%) | 351 (42.5%) | 474 (57.5%) | 718 (87.0%) | 107 (13.0%) |
| | RM9k – 12k | 455 (80.8%) | 108 (19.2%) | 243 (43.2%) | 320 (56.8%) | 482 (85.6%) | 81 (14.4%) |
| | RM12k and above | 477 (78.5%) | 131 (21.5%)*** | 250 (41.1%) | 358 (58.9%)*** | 538 (88.5%) | 70 (11.5%) |
| Knowledge score (t-test) | | 10.4 (1.4) | 10.6 (1.4)** | 10.3 (1.4) | 10.6 (1.4)*** | 10.5 (1.4) | 10.5 (1.4) |

#"Other" includes occupations such as manual labour and contract/ part-time work

*P<0.05

**P<0.01

***P<0.001

Malaysian context. Firstly, the use of face masks is not a norm in Malaysian society. It is uncommon for the typical Malaysian to wear a face mask when ill. The emergence of COVID-19 caused an increase in demand for medical face masks (and hand sanitiser) in the country and supplies were short [23]. The scarcity of face masks meant that many regular members of the public were unable to obtain them. The shortage of personal protective equipment was not limited to Malaysia. It had become a global problem due to increased demand in response to COVID-19 [24]. Secondly, the Ministry of Health Malaysia has been adamant that medical face masks should only be worn by those who are showing symptoms of COVID-19 or similar illnesses. This was to ensure ample supplies of personal protective equipment for medical workers on the frontline. Even so, mixed messages had been communicated to the public by different authoritative bodies on the use of face masks. It is possible that the lack of supply and the confusion caused by the mixed messages led to the divided response on the wearing of face masks when out in public.

Admittedly, COVID-19 has been a teething public health problem around the world. Scientists are working diligently to explore different vaccines and treatment options. Social

scientists, especially those in public health and health communication, are working to identify the levels of knowledge, attitudes and practices on COVID-19 among the public as to design cost-effective public health campaigns and education programmes. The current survey, in fact, exposes the need for more comprehensive education programmes with focus on consistency of information from the government and related authorities. COVID-19 education efforts should take a proactive approach and focus on dispelling misinformation in the form of conflicting opinions, old wives' tales and incorrect information. Due to the levels of media and telecommunication usage in Malaysian society [25–27] and evidence from prior research [28], authorities would benefit from utilising both mainstream and social media in disseminating these messages.

## 5. Limitations

Sampling for the study was conducted via a convenience sample through the networks of the researchers and disseminated through different social media platforms (Whatsapp, Facebook, Twitter etc.). As a result, there is a possibility of bias as underprivileged populations may not have been able to participate in the study. Additionally, when compared to current population statistics in Malaysia [29], the sample of the study were over-representative of women, people below the age of 50, and those employed in the public sector. Therefore, there are limitations to the representativeness of the findings. A more systematic, inclusive sampling method is warranted to improve representativeness and generalisability of the findings.

The second limitation is related to the KAP instrument used in this study. The instrument was adapted from a survey that had been previously tested and utilised in China [13]. Even so, a more thorough assessment of instrument validity and reliability would have produced a more robust instrument. Due to the limited time and urgency of the survey, attitudes and practices were measured with only one item each. In addition to this, possible factors contributing to knowledge, attitude and practice such as risk perceptions and health literacy [30,31] were not measured in this study. These would have been a useful addition in understanding the knowledge, attitudes and practices of COVID-19 in Malaysia.

A further limitation of the present study is the possibility of participants giving socially desirable responses. As this study used self-reported data, it is possible that participants may have answered attitude and practice questions positively based on what they perceive to be expected of them [32].

## 6. Conclusions

In summary, the present study was able to provide a comprehensive examination of the knowledge, attitudes and practices of Malaysians toward COVID-19. The findings suggest that Malaysians have an acceptable level of knowledge on COVID-19 and are generally positive in their outlook on overcoming the pandemic. Even so, consistent messaging from the government and/ or health authorities are key to aid public knowledge and understanding of COVID-19. Additionally, some categories of the population may benefit from specific health education programs to raise COVID-19 knowledge and improve practices.

## Supporting information

**S1 Data.**
(SAV)

## Acknowledgments

We would like to express our appreciation to the Faculty of Social Sciences and Humanities, Universiti Kebangsaan Malaysia, particularly to Associate Professor Dr. Kadaruddin Aiyub and Noraznita Anor Basri for their assistance and a sincere thank you to all members of the Malaysian public who participated in the survey.

## Author Contributions

**Conceptualization:** Arina Anis Azlan, Mohammad Rezal Hamzah, Tham Jen Sern, Suffian Hadi Ayub, Emma Mohamad.

**Data curation:** Arina Anis Azlan, Mohammad Rezal Hamzah, Tham Jen Sern.

**Formal analysis:** Arina Anis Azlan.

**Funding acquisition:** Arina Anis Azlan, Emma Mohamad.

**Investigation:** Arina Anis Azlan, Mohammad Rezal Hamzah, Tham Jen Sern, Suffian Hadi Ayub, Emma Mohamad.

**Methodology:** Arina Anis Azlan, Mohammad Rezal Hamzah, Tham Jen Sern, Suffian Hadi Ayub, Emma Mohamad.

**Project administration:** Emma Mohamad.

**Resources:** Emma Mohamad.

**Supervision:** Emma Mohamad.

**Validation:** Arina Anis Azlan, Mohammad Rezal Hamzah, Tham Jen Sern, Suffian Hadi Ayub, Emma Mohamad.

**Visualization:** Arina Anis Azlan.

**Writing – original draft:** Arina Anis Azlan, Mohammad Rezal Hamzah, Tham Jen Sern, Suffian Hadi Ayub, Emma Mohamad.

**Writing – review & editing:** Arina Anis Azlan, Emma Mohamad.

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
