## [Decision Letter · Decision Letter 0]

7 May 2020

PONE-D-20-11518

Public knowledge, attitudes and practices towards COVID-19: A cross-sectional study in Malaysia

PLOS ONE

Dear Associate Prof. Dr. Mohamad,

Thank you for submitting your manuscript to PLOS ONE. After careful consideration, we feel that it has merit but does not fully meet PLOS ONE’s publication criteria as it currently stands. Therefore, we invite you to submit a revised version of the manuscript that addresses the points raised during the review process.

We would appreciate receiving your revised manuscript by Jun 21 2020 11:59PM. To enhance the reproducibility of your results, we recommend that if applicable you deposit your laboratory protocols in protocols.io, where a protocol can be assigned its own identifier (DOI) such that it can be cited independently in the future. For instructions see: http://journals.plos.org/plosone/s/submission-guidelines#loc-laboratory-protocols

We look forward to receiving your revised manuscript.

Kind regards,

Wen-Jun Tu

Academic Editor

PLOS ONE

Journal Requirements:

Additional Editor Comments (if provided):

Reviewers' comments:

Reviewer's Responses to Questions

**Comments to the Author**

1. Is the manuscript technically sound, and do the data support the conclusions?

Reviewer #1: Yes

2. Has the statistical analysis been performed appropriately and rigorously? 

Reviewer #1: Yes

3. Have the authors made all data underlying the findings in their manuscript fully available?

Reviewer #1: Yes

4. Is the manuscript presented in an intelligible fashion and written in standard English?

Reviewer #1: Yes

5. Review Comments to the Author

Reviewer #1: The authors studied the knowledge levels, attitudes and practices toward COVID-19 among the Malaysian public. 4,850 Malaysian residents was conducted by questionnaire on COVID-19, Most participants held positive attitudes toward the successful control of COVID-19 (83.1%), the ability of Malaysia to conquer the disease (95.9%) and the way the Malaysian government was handling the crisis (89.9%). Most participants were also taking precautions such as avoiding crowds(83.4%) and practising proper hand hygiene (87.8%) in the week before the movement control order started. However, the wearing of face masks was less common (51.2%). So I will give some comments as followings.

The sampling for this study was how to elect ?Will there be deviations in different social media?

How to explain the difference in results between people over 50 and people of other ages

How does income and occupation affect COVID-194.

The following references should be discussed in the revision text.

Cao JL, Hu XR, Tu WJ., & Liu Q. (2020). Clinical Features and Short-term Outcomes of 18 Patients with Corona Virus Disease 2019 in Intensive Care Unit. Intensive Care Medicine, DOI: 10.1007/s00134-020- 05987-7.

Cao JL, Tu WJ, Hu XR, & Liu Q. (2020). Clinical Features and Short-term Outcomes of 102 Patients with Corona Virus Disease 2019 in Wuhan,China. Clinical Infectious Diseases,DOI: 10.1093/cid/ciaa243/ 5814897.

5.What are the reasons for the lack of masks? Any good suggestions?

6. PLOS authors have the option to publish the peer review history of their article (what does this mean?). If published, this will include your full peer review and any attached files.

Reviewer #1: No

---

## [Author Response · Author response to Decision Letter 0]

10 May 2020

Dear Editor,

Thank you for conveying the reviewer’s comments on our article entitled “Public knowledge, attitudes and practices towards COVID-19: A cross-sectional study in Malaysia” (PONE-D-20-11518). We have reviewed the comments and provided more information to address their concerns. This letter will provide you with a point-by-point response as well as details on where the corrections/ additions were made on the manuscript. We have also highlighted the changes in a copy of the manuscript, as requested.

Reviewer report:

Reviewer 1.

1. The sampling for this study was how to elect? Will there be deviations in different social media?

The sampling for this study was based on a convenience sample of professional and personal networks of the researchers. Facebook and Whatsapp were chosen as the two most widely-used communication platforms in Malaysia. The choice to disseminate the survey among different social media applications was to reach participants of different age groups (Pages 5 & 6, lines 126-129).

2. How to explain the difference in results between people over 50 and people of other ages.

The difference in results for people over 50 years of age is addressed in terms of risk perception (Page 15, lines 314-315) and cultural norms (Page 17, lines 356-359) affecting practice in the COVID-19 pandemic.

3. How does income and occupation affect COVID-19.

Income and occupation play a role in knowledge levels as well as attitudes in the COVID-19 pandemic whereby those with lower income may have limited access to timely information on the virus (Page 15, lines 315-317) and those working in the public sector showed higher percentages of positive attitudes toward efforts made in battling COVID-19 (Page 17, lines 344-348). Income also affects practice in that those with higher incomes in terms of compliance to health recommendations and fear and control in pandemic situations (Page 17, lines 359-361).

4. The following references should be discussed in the revision text.

Cao JL, Hu XR, Tu WJ., & Liu Q. (2020). Clinical Features and Short-term Outcomes of 18 Patients with Corona Virus Disease 2019 in Intensive Care Unit. Intensive Care Medicine, DOI: 10.1007/s00134-020- 05987-7.

Cao JL, Tu WJ, Hu XR, & Liu Q. (2020). Clinical Features and Short-term Outcomes of 102 Patients with Corona Virus Disease 2019 in Wuhan, China. Clinical Infectious Diseases, DOI: 10.1093/cid/ciaa243/ 5814897.

The references have been added to the manuscript (Page 3, line 74-75 and page 15, line 314-315) to explain the transmission of the disease and the higher risk levels for older patients.

5. What are the reasons for the lack of masks? Any good suggestions?

The low use of face masks among the Malaysian public was due to societal norms – it is not common for Malaysians to use face masks when they feel sick (Page 17, lines 366-369).

It is also mentioned that the Ministry of Health Malaysia did not encourage the use of medical face masks among the public in order to ensure sufficient supply to healthcare workers (Page 18, lines 372-375).

On the other hand, the lack of face masks available for sale in the country were due to limited global supply (Page 18, lines 370-372).

Thank you very much.

---

## [Editor Report · Decision Letter 1]

12 May 2020

Public knowledge, attitudes and practices towards COVID-19: A cross-sectional study in Malaysia

PONE-D-20-11518R1

Dear Dr. Mohamad,

We are pleased to inform you that your manuscript has been judged scientifically suitable for publication and will be formally accepted for publication once it complies with all outstanding technical requirements.

With kind regards,

Wen-Jun Tu

Academic Editor

PLOS ONE
---

## [Editor Report · Acceptance letter]

15 May 2020

PONE-D-20-11518R1 

Public knowledge, attitudes and practices towards COVID-19: A cross-sectional study in Malaysia 

Dear Dr. Mohamad:

I am pleased to inform you that your manuscript has been deemed suitable for publication in PLOS ONE. Congratulations! Your manuscript is now with our production department. 

With kind regards,

on behalf of

Dr. Wen-Jun Tu 

Academic Editor

PLOS ONE